# Antimicrobial Susceptibility Data for Six Lactic Acid Bacteria Tested against Fifteen Antimicrobials

**Ivana Nikodinoska** [1] , **Jouni Heikkinen** [2] **and Colm A. Moran** [3,*]

1 Alltech European Headquarters, Sarney, Summerhill Road, Dunboyne, A86 X006 Co. Meath, Ireland
2 Biosafe–Biological Safety Solutions Ltd., Microkatu 1M, 70210 Kuopio, Finland
3 Regulatory Affairs Department, Alltech SARL, Rue Charles Amand, 14500 Vire, France
* Correspondence: cmoran@alltech.com

**Abstract:** Antimicrobial resistance is a rising threat in the agrifood sector. The misuse of antibiotics exerts selective pressure, driving resistance mechanisms in bacteria, which could ultimately spread through many routes and render treatments for infectious diseases inefficient in humans and animals. Herein, we report antimicrobial susceptibility data obtained for six lactic acid bacteria, the members of which are commonly used in the food and feed chain. Fifteen antimicrobials were considered for the phenotypic testing: ampicillin, gentamicin, kanamycin, tetracycline, erythromycin, clindamycin, chloramphenicol, streptomycin, vancomycin, quinupristin-dalfopristin, bacitracin, sulfamethoxazole, ciprofloxacin, linezolid, and rifampicin. The reported dataset could be used for the comparison, generation, and reconsideration of new and/or existing cut-off values when considering lactic acid bacteria, particularly lactobacilli and pediococci.

**Keywords:** antimicrobial resistance; lactic acid bacteria; phenotypic testing; *Lactiplantibacillus plantarum*; *Lacticaseibacillus rhamnosus*; *Pediococcus pentosaceus*





## 1. Summary

- Antimicrobial resistance (AMR) in bacteria is one of the significant components of One Health studies due to global concern regarding its impact on public health, food safety, and security. The data reported in this article relate to the antimicrobial susceptibility of six lactic acid bacteria (LAB) representing the species commonly used in the agrifood sector.
- Members of LAB are widespread and are detected in plant materials, foodstuffs, and the guts and mucous membranes of humans and animals. Therefore, the AMR traits of LAB members may be of great importance in describing the AMR distribution within different genera belonging to LAB and within different niches.
- Data generated from the AMR phenotyping analysis of LAB members could enrich the knowledge of LAB behavior when in contact with specific antimicrobials, ultimately being important for the comparison, generation, and reconsideration of new and/or existing cut-off values.

## 2. Data Description

The data reported in this article originated from phenotyping antimicrobial susceptibility testing based on the determination of minimum inhibitory concentrations (MICs) for fifteen antimicrobials. The following six strains were considered in the reported analysis: *Lactiplantibacillus plantarum* (*L. plantarum*) IMI 507026, IMI 507027, and IMI 507028;

*Pediococcus pentosaceus* (*P. pentosaceus*) IMI 507024 and IMI 507025; and *Lacticaseibacillus rhamnosus* (*L. rhamnosus*) IMI 507023. The European Food Safety Authority (EFSA) has reported MIC cut-off values for critically and highly important antimicrobials for use in humans and animals, according to the World Health Organization (WHO), namely ampicillin, gentamicin, kanamycin, tetracycline, erythromycin, clindamycin, chloramphenicol, and streptomycin [1]. In the same guidance document, 'intrinsic resistance' is defined as a common resistance trait against a tested antimicrobial within the same species that does not represent a safety concern; therefore, no cut-off values are reported, nor is testing required. However, when 'acquired resistance' is detected in a strain of a typically susceptible species, with defined cut-off values, this means added risks relating to the antimicrobial resistance pool. To understand whether phenotypical resistance poses a safety risk for horizontal transfer, the whole-genome sequencing-based approach for AMR determinants is proposed [2]. The herein-reported six LAB strains were screened for antimicrobial susceptibility against EFSA-required antimicrobials, as well as against the following non-EFSA required antimicrobials: vancomycin, quinupristin-dalfopristin, bacitracin, sulfamethoxazole, ciprofloxacin, linezolid, and rifampicin.

Table 1 reports antimicrobial susceptibility data collected for six LAB strains isolated from different sources; the triplicate values obtained for each antimicrobial are reported in the publicly available repository Zenodo.

The data obtained for the antibiotic susceptibility of six lactic acid bacteria strains against EFSA-required antimicrobials, as well as the related MIC cut-off values, are shown in Table 1. No official guidelines are available for the non-EFSA-required antimicrobials.

Concerning the antibiotic susceptibility for the six LAB strains against EFSA-required antimicrobials, all six strains showed phenotypic sensitivity to ampicillin, gentamicin, erythromycin, and clindamycin. In addition, the MIC of chloramphenicol exceeded the cut-off values proposed by EFSA for the strains IMI 507023, IMI 507024, and IMI 507025 (MIC: 8 mg/L vs. cut-off value: 4 mg/L). However, the MIC values for chloramphenicol were within one dilution step above the cut-off, being considered to fall within the normal variation of the method [3–10]. Similarly, different feed-related microbial products containing *L. rhamnosus* or *P. pentosaceus* strains showed one dilution step above the cut-off value for chloramphenicol [11–13], but strains within the EFSA cut-off values were also reported [14]. Chloramphenicol is a protein synthesis inhibitor and the presumed acquired resistance toward this antibiotic was correlated with the *cat* (chloramphenicol acetyl transferase) genotype. However, this gene was found in 20 strains out of 79 chloramphenicol-resistant lactobacilli and in 59 strains out of 82 lactobacilli susceptible to this antibiotic [15]. The hypothesis that could explain these inconsistencies include the *cat* position (in the chromosome or plasmid-encoded) [16,17]; the lack of its expression, gene mutation (of the *cat* or genes related to *cat* regulatory mechanism) [17]; or the possible involvement of different mechanisms such as low expression of different genes, e.g., genes encoding outer membrane proteins [18]. The whole-genome sequencing of the six LAB strains herein reported was screened against different available antimicrobial resistance databases and no hits were found [4–9]. In addition, the plasmid-related genes were not detected, suggesting no risk of horizontal AMR transfer or any raised safety concerns.

The two pediococci strains, IMI 507024 and IMI 507025, exceeded the cut-off value by one dilution for kanamycin (MIC: 128 mg/L vs. cut-off value 64 mg/L), whereas one out of the six strains, IMI 507024, exceeded the cut-off value by one dilution for streptomycin (MIC 128 mg/L vs. cut-off value: 64 mg/L), thus posing no safety concern [5,6]. Higher phenotypical resistance towards aminoglycosides is widely reported in the literature [15,18–20]. Different studies reported higher kanamycin MIC values for pediococci: 22 out of 35 strains were between 64–128 mg/L [21], 4 out of 6 pediococci strains were found to be kanamycin- and streptomycin-resistant [18], and 20 out of 28 strains showed values of 256 mg/L or higher [22]. Genotype–phenotype correlation was mostly investigated for lactobacilli; genes encoding kanamycin and streptomycin kinases *aph(3″)-IIIa*, *str(A)/str(B)* or adenyltransferase genes (*aadA*, *aadE*) did not consistently explain the phenotypic resistance towards

these antimicrobials, suggesting that possible intrinsic resistance though the enzymatic antibiotic inactivation could not be excluded [18,19]. The potential intrinsic resistance nature toward aminoglycosides, mostly kanamycin and streptomycin, was suggested to be linked with limited or a lack of cytochrome-mediated drug transport [20,23,24].

**Table 1.** Minimum inhibitory concentration (MIC) values for six lactic acid bacteria against 15 antimicrobials. The quality control strains used were *Lacticaseibacillus paracasei* LMG12586 (ATCC 334) and *Streptococcus pneumoniae* DSM 24048 (ATCC 49619).

| Strain | Ampicillin | Gentamicin | Kanamycin | Tetracycline | Erythromycin | Clindamycin | Chloramphenicol | Streptomycin |
|---|---|---|---|---|---|---|---|---|
| | **Antibiotic Susceptibility (MIC mg/L)—EFSA required antimicrobials** | | | | | | | |
| *L. plantarum* IMI 507026 | 0.25 | 1 | 32 | 32 | 0.25 | ≤0.03 | 8 | 8 |
| *L. plantarum* IMI 507027 | 0.25 | 1 | 64 | 32 | 0.25 | ≤0.03 | 8 | 16 |
| *L. plantarum* IMI 507028 | 0.25 | 1 | 32 | 32 | 0.25 | ≤0.03 | 8 | 4 |
| *L. rhamnosus* IMI 507023 | 1 | 1 | 16 | 1 | 0.12 | 0.5 | 8 | 2 |
| *P. pentosaceus* IMI 507024 | 4 | 4 | 128 | 64 | 0.5 | 0.06 | 8 | 128 |
| *P. pentosaceus* IMI 507025 | 4 | 4 | 128 | 64 | 0.5 | 0.06 | 8 | 64 |
| *L. paracasei* LMG12586 (ATCC 334) | 1 | 1 | 32 | 2 | 0.12 | 0.06 | 4 | 16 |
| **Strain** | **Microbiological MIC cut-off values (mg/L)** | | | | | | | |
| *L. plantarum* | 2 | 16 | 64 | 32 | 1 | 4 | 8 | - |
| *L. rhamnosus* | 4 | 16 | 64 | 8 | 1 | 4 | 4 | 32 |
| *P. pentosaceus* | 4 | 16 | 64 | 8 | 1 | 1 | 4 | 64 |
| *L. paracasei* LMG12586 (ATCC 334) | 0.5–2 | 1–8 | 8–64 | 1–4 | 0.03–0.5 | 0.06–0.25 | 4–8 | 4–32 |

| Strain | Vancomycin | Quinupristin-dalfopristin | Bacitracin | Sulfamethoxazole | Ciprofloxacin | Linezolid | Rifampicin |
|---|---|---|---|---|---|---|---|
| | **Antibiotic susceptibility (MIC mg/L)—non-EFSA required antimicrobials** | | | | | | |
| *L. plantarum* IMI 507026 | >128 | 1 | 256 | 64 | 16 | 2 | 4 |
| *L. plantarum* IMI 507027 | >128 | 1 | 128 | 256 | 16 | 2 | 8 |
| *L. plantarum* IMI 507028 | >128 | 1 | 256 | 256 | 16 | 2 | 8 |
| *L. rhamnosus* IMI 507023 | >128 | 1 | 32 | >512 | 2 | 2 | ≤0.12 |
| *P. pentosaceus* IMI 507024 | >128 | 2 | 64 | >512 | 32 | 4 | 2 |
| *P. pentosaceus* IMI 507025 | >128 | 2 | 16 | >512 | 16 | 4 | 2 |
| *S. pneumoniae* ATCC 49619 (mg/L) | ≤0.25 | 1 | 32 | >512 | 2 | 1 | ≤0.12 |
| Antimicrobial plate range (mg/L) | 0.25–128 | 0.015–8 | 1–512 | 1–512 | 0.004–128 * | 0.03–16 | 0.12–64 |

\* Range on FINBIOS1 test plate (0.004–0.25) and range on EULACBI1 test plate (0.25–128).

sensitive     resistant     no cut-off values are available

The MIC observed for the tetracycline exceeded the cut-off value by more than one dilution (MIC 64 mg/L vs. cut-off value 8 mg/L) for only the two *Pediococcus pentosaceus* strains IMI 507024 and IMI 507025. In contrast, the *L. plantarum* and *L. rhamnosus* strains showed values under the proposed microbial cut-off. Feed additives composed of pediococci strains were found to exceed the cut-off values for one dilution for tetracycline, considered as no risk [13,25] or sensitive towards this antibiotic [26,27]. In a recent study,

34 out of 35 pediococci strains showed MIC values from 32–128 mg/L, and no genetic determinants that could explain these resistances were found when screening the whole-genome sequencing using the AMRFinderPlus, CARD, ARG-ANNOT v.4, and Resfinder v.3.0 AMR databases [21]. The most reported determinants for tetracycline resistance were genes encoding for energy-dependent efflux systems (*tetA, tetB, tetC, tetE*), ribosomal protection proteins (*tetM, tetO, tetS*), or tetracycline degradation enzyme (*tetX*) [28]. Although the most reported genetic determinants for tetracycline-acquired resistance in lactobacilli were *tetL, tetM, tetP, tetQ, tet S, tetW* [15], similar genetic determinants were not consistently detected in phenotypic resistant pediococci strains, suggesting potential species-specific intrinsic resistance toward this antibiotic [21,28,29]. The potential intrinsic resistance nature was hypothesized to be linked with the complex network that regulates antibiotic uptake [18].

Among protein synthesis inhibitor antimicrobials, MIC cut-off values were proposed for gentamicin, kanamycin, tetracycline, streptomycin, erythromycin, clindamycin, and chloramphenicol, but not for quinupristin-dalfopristin and linezolid. Concerning quinupristin-dalfopristin, an MIC of 4 mg/L was proposed for lactobacilli and pediococci [22]. The MIC screening of 182 lactobacilli showed that 0.5–4 mg/L in the MIC range is more frequently observed for quinupristin-dalfopristin, whereas only 10 strains showed MIC values of 8 and 16 mg/L [15]. Similarly, the proposed MIC value for linezolid is 4 mg/L [22,29], being reported between 1–4 mg/L for MIC values for lactobacilli [15] and between 1–6 mg/L for pediococci [29]. The strains herein reported ranged from 1–2 mg/L and 2–4 mg/L, suggesting their quinupristin-dalfopristin and linezolid sensitivity, respectively.

Three cell wall synthesis inhibitors were evaluated in the present study: ampicillin, vancomycin, and bacitracin. All strains were sensitive to ampicillin, according to the EFSA cut-off values, whereas the vancomycin intrinsic resistance was confirmed by the consistently high MIC values for all strains, being >128 mg/L. The vancomycin resistance in LAB is due to the synthesis and incorporation of peptidoglycan precursors containing D-Lac instead of D-Ala, resulting in decreased vancomycin affinity [15,30]. MIC values for bacitracin are scarcely reported in the literature; 11 out of 18 *L. plantarum* strains, 12 out of 22 *L. paracasei/rhamnosus*, and 3 out of 6 *L. sakei/curvatus* showed MIC values of ≥256 mg/L, whereas the MIC values for the *L. acidophilus* group were mostly in a range between 2 and 32 mg/L [20]. These potential ranges were in accordance with the observations in the present study.

MIC values for nucleic acid inhibitors, sulfamethoxazole, ciprofloxacin, and rifampicin are not covered in the EFSA guidelines. Different authors stressed potential intrinsic resistance towards this antimicrobial group due to high MIC values, potentially being related to the structure of the cell wall and the impermeability of the membrane [22,31]. High MIC values were reported for sulfamethoxazole, being 64–1024 mg/L for different LAB, and 512 was proposed as a breakpoint (µg/mL) [32,33]. The MIC value of >32 mg/L was proposed for ciprofloxacin for lactobacilli and pediococci [20,29]. The MIC values for rifampicin ranged between 0.12–4 mg/L [15,20,21]. These potential ranges were in accordance with MIC values for the six strains reported in the present study, being 64->512 mg/L and 2–32 mg/L, for sulfamethoxazole and ciprofloxacin, respectively. Most of the strains were within the range observed in the literature for rifampicin, except for the two *L. plantarum* strains, IMI 507027 and IMI 507028, which showed one dilution higher than the observed MIC range, being considered within the normal variation of the method [3].

The six LAB strains herein presented were previously sequenced and screened against four antimicrobial resistance databases (ARG-ANNOT, MEGARes, NCBI Bacterial Antimicrobial Resistance ReferenceGene Database, and ResFinder) to assess their antimicrobial resistance potential via the whole-genome sequencing approach. No genes were found that correlate with any antimicrobial resistance genes [34–39].

In conclusion, the antimicrobial susceptibility data for the strains *Lactiplantibacillus plantarum* (*L. plantarum*) IMI 507026, IMI 507027, and IMI 507028; *Pediococcus pentosaceus* (*P. pentosaceus*)

IMI 507024 and IMI 507025; and *Lacticaseibacillus rhamnosus* (*L. rhamnosus*) IMI 507023 raised no antimicrobial resistance safety concerns [4–10].

## 3. Methods (Required)

### 3.1. Microbial Strains and Origin

The features of the six lactic acid bacteria strains, including their niche origins and genome accession numbers, are reported in Table 2.

**Table 2.** Lactic acid bacteria included in the phenotypical antimicrobial susceptibility screening.

| Species | Strain | Isolation Source | GenBank Accession Number | Reference |
|---|---|---|---|---|
| *Lacticaseibacillus rhamnosus* | IMI 507023 | Corn silage | JAJBZL000000000 | [34] |
| *Pediococcus pentosaceus* | IMI 507024 | Fermented sausage | JAJFOC000000000 | [35] |
| *Pediococcus pentosaceus* | IMI 507025 | Pickled cucumbers | JALBYI000000000 | [36] |
| *Lactiplantibacillus plantarum* | IMI 507026 | Corn silage | JALBGE000000000 | [37] |
| *Lactiplantibacillus plantarum* | IMI 507027 | Grass silage | JAJTVG000000000 | [38] |
| *Lactiplantibacillus plantarum* | IMI 507028 | Fermented milk | JAKMAX000000000 | [39] |

### 3.2. Test Material

The freeze-dried pure culture for the six test strains IMI 507023, IMI 507024, IMI 507025, IMI 507026, IMI 507027, and IMI 507028 was deposited at the Centre for Agriculture and Bioscience International (CABI) Culture Collection. For the antimicrobial susceptibility analysis, freeze-dried cultures provided by CABI were considered.

### 3.3. Antimicrobial Susceptibility Assay

The antibiotic susceptibility of the strains to ampicillin, gentamicin, kanamycin, tetracycline, erythromycin, clindamycin, and chloramphenicol was analyzed according to the ISO10932:2010 standard with the VetMIC Lact-1 and VetMIC Lact-2 plates (SVA National Veterinary Institute, Uppsala, Sweden) in anaerobic conditions at $+37 \pm 1$ °C for 48 h using LAB susceptibility test medium (LSM). *Lacticaseibacillus paracasei* (*L. paracasei*) LMG12586 (ATCC 334) was used as a quality control strain. The data for minimum inhibitory concentrations were obtained through the visual inspection of the growth utilizing magnifying reading mirror (SVA National Veterinary Institute, Uppsala, Sweden). The strains were grown on MRS agar species-specific conditions for 48 h. The A625 was adjusted to 0.18 ($\pm$0.02) and further diluted in LSM broth to reach $3 \times 10^4$ cfu/well for the test.

The antimicrobial susceptibility against streptomycin, vancomycin, quinupristin-dalfopristin, bacitracin, sulfamethoxazole, ciprofloxacin, linezolid, and rifampicin was analyzed using the broth microdilution method according to the CLSI standard M07 (11th ed., 2018) 'Methods for dilution antimicrobial susceptibility tests for bacteria that grow aerobically', which describes the general method, except that custom-made microdilution trays were used. The CLSI standard M45 (3rd ed., 2016) 'Methods for antimicrobial dilution and disk susceptibility testing of infrequently isolated or fastidious bacteria' for *Lactobacillus* spp. was followed in the selection of the medium and incubation conditions. The strains were grown at species-specific conditions for 48 h. Based on a preliminary study, the A550 was adjusted to 0.125 ($\pm$0.01), corresponding to approximately 1.3–6.6 $\times 10^7$ cfu/mL depending on the strain. The cell suspension was further diluted in broth to reach $5 \times 10^4$ cfu/well for the test. The test was performed in triplicate with custom-made Sensititre™ FINBIOS1 and EULACBI2 96-well plates (Thermo Scientific), as well as with plates prepared in house (bacitracin, sulfamethoxazole) according to the CLSI standard M07, 11th ed., under a 5% $CO_2$ atmosphere at $35 \pm 2$ °C, for $48 \pm 2$ h using cation-adjusted Mueller Hinton Broth containing 5% lysed horse blood (Oxoid, CM0405, cations $Mg_2^+$ and $Ca_2^+$ added separately). *Streptococcus pneumoniae* DSM 24048 (ATCC 49619) was used as a quality control strain and was incubated for $22 \pm 2$ h.

**Author Contributions:** I.N.: writing—original draft and data curation; J.H.: writing—review and editing, methodology, and formal analysis; C.A.M.: writing—review and editing, supervision, and project administration. All authors have read and agreed to the published version of the manuscript.

**Funding:** This work was sponsored by Alltech SARL (France).

**Institutional Review Board Statement:** Not applicable.

**Informed Consent Statement:** Not applicable.

**Data Availability Statement:** The raw data obtained for each antimicrobial are reported in publicly available repository Zenodo: https://doi.org/10.5281/zenodo.6826193.

**Conflicts of Interest:** The authors I.N. and C.A.M. are employees of Alltech, which produces the lactic acid bacteria evaluated in this study.

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
