# Peer review of "Antimicrobial Susceptibility Data for Six Lactic Acid Bacteria Tested against Fifteen Antimicrobials"

_data_

Round 1
Reviewer 1 Report
Research Description:
This study describes the antimicrobial susceptibility of six lactic acid bacteria to 15 antibiotics.
Major critics:
The study is limited to only 6 lactic acid bacteria and comparison to other reference strains in the database should be considered. For example, compare the results to strains of lactic acid bacteria that are already being marketed as probiotics in various foods.
Abstract: correct the errors; the antibiotic names are given twice (lines 15-16).
Methods: There is a lack of standards in the field of susceptibility testing and relative definition of breakpoints for different species of lactobacilli, therefore the parameters that might interfere with the antibiotic profile should be considered. Since LSM medium is not yet considered the standard medium for MIC assessment, did the authors compare the results to lactobacilli grown on MRS medium? Are there similar results?
Results: The intrinsic resistance of lactic acid bacteria to kanamycin and tetracycline has already been mentioned in the literature. Furthermore, vancomycin-resistant LAB strains are believed to be intrinsic due to D-Ala-d-lactate in their peptidoglycan instead of the natural dipeptide D-Ala-D-Ala. Additional genomic data would explain these points.
Other papers provide data on tetracycline MIC cut-off values ​​compared to EFSA and CLSI reference values. The target strains' genotype (tetracycline gene) should be included along with the antibiotic profile.
The results must be discussed in comparison with other similar studies. There is missing literature to be cited.
Author Response
Dear Reviewer,
Thank you for taking the time to review our manuscript. We have taken your comments and feedback into consideration and responded in the attached file. The manuscript is more substantial after your review.
Kind regards,
The authors

Reviewer 2 Report
The authors conducted drug susceptibility tests on six strains of lactic acid bacteria isolated from food products. This manuscript seems to be a significant study because it is possible to determine what kind of selection pressure is applied to these strains by examining the drug sensitivity of these lactic acid bacteria with which we are in daily contact.
L74-&L79-
Please indicate the basis for your conclusion that " the MIC values for chloramphenicol were within one dilution step above the cut-off and thus did not raise safety concerns" from the EFSA guidelines you cited. If you cannot show this, then this conclusion is unacceptable.
After stating in the Abstract that 15 antimicrobial resistances were tested, 20 antimicrobials are listed. The 5 antimicrobials that have not been tested or for which no data can be presented should not be listed.
L51
Since the text cites the definitions of intrinsic and acquired resistance, it would be necessary to indicate whether each of the 15 antimicrobial resistances listed in Table 1 is intrinsic or acquired resistance for each bacterial species (or strains). Is it correct to understand that the 8 drugs in Table 1A for which EFSA has set cutoff values are acquired resistant and the 7 drugs in Table 1B for which no cutoff values have been set are intrinsically resistant? If so, it should be clearly stated in the text and the legend.
Based on the above, there is no reference to Table 1B in the text, and Table 1B needs to be explained in the text.
There is no Table2 and there is a Table3.
“Silage” in Table3 is capitalized and lowercased. Please unify them.
It should be considered whether or not the current situation is becoming more or less tolerant, and the difference in the results depending on the species.
Author Response
Dear Reviewer,
We are grateful for the time and effort you have made to review our manuscript. We believe the manuscript is improved based on your comments.
thank you
The authors

Reviewer 3 Report
Dear Authors,
The work that was shown to us was quite intriguing. Antimicrobial susceptibility data which are thought to be topped with six lactic acid bacteria against fifteen microbials, are the most appealing of these types of antimicrobial susceptibility data. Work is succinct, and its effects are readily apparent. Nonetheless, it is a fresh take on the topic that adds to our understanding of the process. A visual representation of the antimicrobial susceptibility data for six lactic acid bacteria tested against fifteen antimicrobials passivated with the prospective data might be helpful to readers. I hope the best for the writers and strongly suggest they have this piece published.
Author Response
Dear reviewer,
We are grateful for your time and expertise in reviewing our manuscript.
Thank you for your positive feedback and we look forward to the work being published.
Kind regards,
The authors.
Round 2
Reviewer 1 Report
The genes should be italicized throughout the text (lines: 113-116).
The reference at line 109 is missing.
Please check all the references.
Author Response
Dear Reviewer,
Thank you once again for your time and expertise.
The manuscript was amended accordingly.
Kind regards,
The authors
Reviewer 2 Report
>Q12: It should be considered whether or not the current situation is becoming more or less tolerant, and the difference in the results depending on the species.
>A12: We cannot understand what is referred to. However, the sensitivity towards antimicrobials was described in the revised version of the manuscript.
The paper would be more valuable if it not only presents the results, but also discusses possible problems with the results and why such results were obtained.
Author Response
Dear Reviewer,
Thank you for your expert review of our manuscript. We are grateful for the precision in response to our question.
We want to provide context regarding the limited response we have prepared. As per the Aim & Scope of the journal DATA (https://www.mdpi.com/journal/data/about), our manuscript aims to enhance data transparency and to provide a dataset so that researchers in this field could include in further work and work together in outlining the most reliable cut-off values for this microbial group. Therefore, our manuscript as Data Descriptors (https://www.mdpi.com/journal/data/about) describes a single dataset collected for phenotypic antimicrobial susceptibility analysis.
However, we agree with your comment, so we have briefly discussed the findings with other published work in the manuscript. Concerning genotypic-phenotypic comparison, as mentioned in the manuscript, inconsistencies and some hypothetical reasons collected from the literature are included in our manuscript. However, our WGS analysis findings in the text align with the literature observations, as no resistance genes were detected. Therefore, we could not confirm the phenotypic resistances found.
We appreciate your advice in investigating further and look forward to addressing these fundamental questions through our future investigations.
Thank you
The Authors